Dazzling damselfish: investigating motion dazzle as a defence strategy in humbug damselfish (Dascyllus aruanus)

Tosetto Louise louise.tosetto@mq.edu.au
Hart Nathan S.
Ryan Laura A.
School of Natural Sciences, Macquarie University , NSW , Australia
Kramer Donald
Electronic publication date: 2024 Sep 25
Publication date: 2024
Volume: 12
Electronic Location ID: e18152
Received 2024 Jan 18; Accepted 2024 Aug 31
Copyright: ©2024 Tosetto et al.
Copyright year: 2024
Copyright holder: Tosetto et al.
License: This is an open access article distributed under the terms of the Creative Commons Attribution License, which permits unrestricted use, distribution, reproduction and adaptation in any medium and for any purpose provided that it is properly attributed. For attribution, the original author(s), title, publication source (PeerJ) and either DOI or URL of the article must be cited.
License URL: https://creativecommons.org/licenses/by/4.0/

Keywords: Camouflage, Animal colouration, Antipredator strategy, Behaviour, Vision

Funding: School of Natural Sciences, Macquarie University; ARC Linkage Project LP1900099 Macquarie University Research Fellowship The project was supported by the School of Natural Sciences (formerly the Department of Biological Sciences) at Macquarie University, ARC Linkage LP19000992 and Macquarie University Research Fellowship (Tosetto). The funders had no role in study design, data collection and analysis, decision to publish, or preparation of the manuscript.

==============================
Many animals possess high-contrast body patterns. When moving, these patterns may create confusing or conflicting visual cues that affect a predator’s ability to visually target or capture them, a phenomenon called motion dazzle. The dazzle patterns may generate different forms of optical illusion that can mislead observers about the shape, speed, trajectory and range of the animal. Moreover, it is possible that the disruptive visual effects of the high contrast body patterns can be enhanced when moving against a high contrast background. In this study, we used the humbug damselfish (Dascyllus aruanus) to model the apparent motion cues of its high contrast body stripes against high contrast background gratings of different widths and orientations, from the perspective of a predator. We found with higher frequency gratings, when the background is indiscriminable to a viewer, that the humbugs may rely on the confusing motion cues created by internal stripes. With lower frequency gratings, where the background is likely perceivable by a viewer, the humbugs can rely more on confusing motion cues induced by disruption of edges from both the background and body patterning. We also assessed whether humbugs altered their behaviour in response to different backgrounds. Humbugs remained closer and moved less overall in response to backgrounds with a spatial structure similar to their own striped body pattern, possibly to stay camouflaged against the background and thus avoid revealing themselves to potential predators. At backgrounds with higher frequency gratings, humbugs moved more which may represent a greater reliance on the internal contrast of the fish’s striped body pattern to generate motion dazzle. It is possible that the humbug stripes provide multiple protective strategies depending on the context and that the fish may alter their behaviour depending on the background to maximise their protection.

Introduction

Avoiding detection is key to survival for many animals and there are several tactics that animals may use to decrease detection. Visual camouflage, chemical concealment and modification of vocalisations (see Ruxton, 2009 and examples within) are all strategies that animals may use to maintain crypsis. However, visual camouflage is the most understood phenomenon with strategies such as the alteration of colouration or patterning, body positioning or self-shadowing all assisting an animal to blend into its environment. Many of these camouflage strategies are only thought to be effective when prey is still, with movement likely to increase conspicuousness and provide locational information to predators (Tan & Elgar, 2021). While avoiding detection is arguably the most effective form of predator defence, remaining still is not always practical. Once an animal breaks camouflage and is identified, avoiding capture is often achieved by deceiving or confusing predators (Tan et al., 2024).

Strategies that confuse predators incorporate patterning and behaviour to creating concealing or misleading motion signals (see Tan et al., 2024). Concealing motion signals, such as flicker-fusion, allow prey to exploit a predator’s visual limitations. When prey move fast enough there is an apparent blurring of body patterning, making them appear less conspicuous (Umeton et al., 2019; Valkonen et al., 2020). Misleading motion signals, such as motion dazzle, are thought to be generated through contrasted stripes or zig-zag patterning. When these patterned animals move, they can hinder a predators perception of their trajetory, speed and range (Scott-Samuel et al., 2023). To date there is little empirical evidence of the use of these strategies in nature (Tan et al., 2024). However, several studies have used human observers to demonstrate that motion dazzle can hinder the ability to accurately capture targets (Stevens, Yule & Ruxton, 2008), perceive the speed of targets (Hall et al., 2016; Kodandaramaiah et al., 2020) and judge direction of targets (Hughes et al., 2017). Further validation to the concept of motion dazzle has been obtained by taking a comparative phylogenetic approach in snake and lizard patterning. In snakes, a correlation was found between the presence of longitudinal stripes (parallel to the body length) and small, fast, exposed snakes, suggesting contrasting body patterns function efficiently during movement (Allen et al., 2013). Similarly, a phylogenetic approach found that conspicuously striped lizards were substantially more mobile than cryptic lizards, indicating that the striping may enhance escape strategies via motion dazzle (Halperin, Carmel & Hawlena, 2017). Furthermore, using comparative methods and eco-physiological factors, it was found that lizards with longitudinal striped tails are likely ground dwelling, have higher body temperatures, are (mostly) diurnally active, and can lose their tail, supporting the notion that striped tails in lizards may have protective functions based on motion dazzle effects (Murali, Merilaita & Kodandaramaiah, 2018).

Animal body coloration and patterning may not be constrained to one defensive role but rather offer multiple protective strategies. It has been suggested that disruptive patterns may be combined with warning colours in apparently conspicuous signals (Stevens & Merilaita, 2009). For instance, the highly conspicuous patterning of poison dart frogs (Dendrobatidae), helps them blend into the background when viewed from a distance, but once detected, the same markings provide an aposematic function (Barnett et al., 2018). Similarly, given that high contrast patterns such as stripes and zig-zags are effective in providing both disruptive and dazzle colouration (Scott-Samuel et al., 2023), it is likely there is a dual function to this patterning. Animals with highly contrasting body patterns may benefit from background matching and disruptive camouflage while static. But when moving, these same patterns may provide protection through aposematism, mimicry and confusing motion signals. This has been demonstrated in European vipers (genus Vipera), small, highly motile reptiles, with contrasted zig-zag patterning. The patterning which serves to provide crypsis when curled and still, can also hinder the probability of capture through motion dazzle when moving (Valkonen et al., 2020). To date, examples of these multiple defence strategies have not been well documented and it is likely that using body patterning for different protective tactics extends into the aquatic environment as well.

It is possible that the high-contrast patterning of the humbug damselfish (Dascyllus aruanus) provides different protective roles depending on the context. Humbugs are a small (10 cm total length TL) black and white species of coral reef fish. They have three black transverse stripes (perpendicular to their body), a white spot between the eyes and a white tail (Kuiter, 1996) (Fig. 1). Humbugs inhabit branching Acropora sp. and Pocillopora sp. coral colonies and have been shown to seek refuge in the complex architecture of the branching corals (Holbrook & Schmitt, 2002). A recent study demonstrated that humbug damselfish (hereafter ‘humbugs’) may receive protection on the reef using disruptive camouflage (Phillips et al., 2017). Using static humbug models, Phillips et al. (2017) found that the greatest camouflage was attained when the backgrounds were of similar or slightly smaller width (high spatial frequency) to the humbug stripes and when the humbug stripes were orientated like the background. However, humbugs are rarely static and regularly move around coral patches in restricted-entry social groups of between 2 and 25 individuals (Holbrook, Forrester & Schmitt, 2000; Mann et al., 2014). Given their high contrast striped body, it is possible that these fish are also using motion dazzle to confuse predators when moving. Furthermore, motion dazzle is reportedly more common in small species, which are highly mobile and can inhabit open spaces (Scott-Samuel et al., 2023). While humbugs do not always inhabit open-water environments, they do move within and between coral colonies (Mann et al., 2014). Moreover, they will prioritize feeding outside of the protective coral colony during high tide when plankton is readily available (Kent et al., 2019). In this regard, the humbug provides an excellent model to explore whether contrasted stripes may be providing an additional protective strategy via motion dazzle.

Figure 1 Banded humbug damselfish in branching coral habitat.

Banded humbug damselfish, Dascyllus aruanus within branching corals. Image source credit: have seen from iStock (https://www.istockphoto.com/photo/coral-reef-at-maldives-gm483076145-37362062?clarity=false).

A previous study demonstrated the mechanics of how high contrast patterns generate misleading and confusing information by applying a simulated biological visual system based on a two-dimensional motion detection (2DMD) algorithm to high contrast stripes of zebras (How & Zanker, 2014). How & Zanker (2014) found that zebra movement created confusing motion signals including motion opposing the direction in which the animal was moving. It is likely that movement of humbugs creates the same confusion to motion signals. In addition, different contrasted backgrounds have been shown to affect detectability, and motion dazzle in zebras may be enhanced when moving with other zebras in a herd (How & Zanker, 2014). In this regard, there may be even more protection for humbugs when they move against similarly contrasted backgrounds, such as corals or other humbugs (Dimitrova et al., 2009). It is possible that the motion from the humbug stripes, combined with motion from the background, which is also likely interrupted at the humbug body edges will provide even greater motion dazzle. Further, some animals can assess their degree of camouflage and predation risk and adjust behaviour to maximise camouflage (Kang et al., 2015; Wilson-Aggarwal et al., 2016). Given evidence of disruptive camouflage against backgrounds of similar spatial frequencies, and that motion dazzle may also be enhanced against different backgrounds, we wanted to explore whether humbugs modify behaviour with different backgrounds in a way which may reduce detection.

In this study, we first built on the work conducted by How & Zanker (2014) by using the same motion model that was applied to zebra patterning. We extended this to assess whether high contrast backgrounds interact with the humbug pattern to enhance motion dazzle and whether motion cues may also be interrupted at the edges of animals, particularly against highly contrasted backgrounds. By using uniform square-wave gratings at different spatial frequencies (i.e., grating width) and orientations we explored baseline differences obtained by different spatial frequencies without the complexities found in the natural environment. We predicted that those backgrounds with similar frequency and orientation to that of the humbugs would provide the greatest disruption to motion cues and be most effective in reducing detectability from the perspective of a moving predator. We then explored whether humbugs could perceive different background spatial patterns and modify their behaviour to minimise detectability. We examined the humbug eye to establish whether humbugs have the visual acuity to perceive differences in backgrounds of varying spatial frequencies. We then assessed whether humbugs spent more time closer, or moved more in response to different backgrounds. Backgrounds with spatial frequencies similar or slightly higher to humbug stripes, and that are orientated in similar direction (Phillips et al., 2017) provide humbugs with the greatest disruptive camouflage. Thus, we hypothesised that humbugs would spend more time closer, and move more in response to those backgrounds that are more effective in reducing detectability. Given that width of the humbug stripes was around 0.5 cm, we predicted the grating best at reducing detectability would be 0.5 cm with a vertical orientation.

Materials & Methods

Animal acquisition and housing

All procedures were approved by the Macquarie University Animal Ethics Committee (ARA 2017/039).

Ten wild-caught humbugs were obtained from a local aquarium supplier in Sydney, Australia and transported in aerated seawater to the Macquarie University Seawater Facility. This facility comprises a total of 45,000 L of recirculated seawater, which is collected from depth in Sydney Harbour. Humbugs were housed individually in opaque white polythene tubs (600 × 350 × 400 mm) and maintained at a water flow rate of 3 L min−1, salinity of 35 ppt and temperature of 26 °C. Aquaria were illuminated with aquarium LED lights (Aqua One Strip Glo Marine 90 cm Aquarium LED Light) on a 12:12 h light:dark regime. A white PVC pipe (100 mm long, 100 mm diameter) in each tank served as a shelter for the fish. Once a day, aquaria were cleaned and fish were fed to satiation with Nutridiet marine flakes (Seachem, Madison US). Fish were acclimated to the aquaria for two weeks before trials commenced. All procedures were approved by the Macquarie University Animal Ethics Committee (ARA 2017/039).

Test backgrounds/gratings

We created several different square-wave test gratings to use as backgrounds for the motion dazzle modelling and subsequent behavioural trials. Gratings were created in Adobe Illustrator (version 22.1, 2018) and consisted of repeating black (RGB: 0,0,0) and white (RGB: 255,255,255) bars of equal width. Five different spatial frequencies were generated where the widths of the individual grating ‘bars’ (black or white) were 1 cm, 0.5 cm, 0.25 cm, 0.1 cm and 0.05 cm (see Fig. S1 for examples of the gratings). These gratings cover the range of spatial frequencies of Acropora branching corals in which the humbugs are usually found (Phillips et al., 2017). Cards were printed on 250GSM A3 matte photographic paper (Krisp, Hoppers Crossing VIC, Australia). Background cards were cut to cover the test board (22 × 20 cm) and laminated as per Phillips et al. (2017). Transparent plastic pouches (gloss, A3, 125 µm thick; GBC® Signature laminating pouches; GBC, Lake Zurich, IL, USA) were used to laminate the background test cards. Although we use unnatural striped backgrounds, these results can provide insight into the potential dazzle that high contrast body patterns may offer, particularly in a highly complex and structured visual environment and/or for animals that form groups.

Estimating the motion dazzle effect

We recorded video footage of three humbugs as they swam against different backgrounds from the perspective of a moving predator. This was performed in a 40 × 22 cm aquarium with the test board and background (22 × 20 cm) placed at one end of the aquarium, and the camera positioned 30 cm away from the background at the opposing end. The remaining walls of the aquarium were opaque grey, and the aquarium was lit from above (120 cm DEE Full Spectrum Marine Aquarium LED Light); thus, there were no distortions due to light reflections. A GoPro™ Hero 9 video camera was attached to a camera dolly and pulled along a track fixed parallel to the grating so that the movement of the camera from one side of the arena to the other was smooth, level, and at a consistent speed. We moved the GoPro from side-to-side at ∼15–20 cm s−1. Fish would stay close to the grating (<5  cm) as it was the farthest position in the tank from the moving camera; thus, fish were consistently recorded against background gratings. We used background gratings of three different spatial frequencies (1 cm, 0.25 cm and 0.05 cm) which were presented in both vertical and horizontal orientation. These frequencies were used as they represent grating widths that could not be discriminated (0.05 cm), are close to the limit (0.25 cm) and could be easily discriminated by a typical predator (1 cm). Using these three grating sizes we could deduce the results of a broader range of grating widths. In total, three replications of each fish were recorded against each of the six different backgrounds.

The motion cues created by the humbug were analysed with respect to the visual abilities of a potential predator, the slingjaw wrasse (Epibulus insidiator) or the coral trout (Plectropomus leopardus). The analysis was performed on a 600 × 600-pixel region of interest (ROI) extracted from each video frame, starting from when the humbug entered and finishing when it exited this grid (ranging from 8 to 23 frames for a video). Videos were cropped to ensure the motion of the humbug was consistent and the dominant feature. We also analysed the same video clips with the ROI positioned so that it only contained the background grating, thus calculating the motion that was created due to the movement of the camera (i.e., the potential predator). All videos were analysed so that the humbug moved from the left to the right of the screen. For some videos this was achieved by flipping the video horizontally.

Visual motion at the level of the retina was estimated from the videos using a 2-dimensional motion detection (2DMD) model (How & Zanker, 2014; Pallus, Fleishman & Castonguay, 2010) written in MATLAB.

The 2DMD model uses two orthogonal arrays of elementary motion detectors (Reichardt, 1987) to compare pixel intensities at a set pixel spacing and between frames based on a set temporal and spatial filter. This model is a simplified version of the motion detecting circuitry present in a wide range of animals (Barlow & Levick, 1965; Borst, Haag & Reiff, 2010; Takemura et al., 2011). The correlation of pixel intensities based on spatiotemporal filters is used to determine the strength and direction of object motion. The frame rate of the videos (30 frames per second, i.e., 30 Hz) is similar to the temporal resolution thresholds in fishes (Fritsches, Brill & Warrant, 2005; Matsumoto et al., 2009; Pallus, Fleishman & Castonguay, 2010; Pusch et al., 2013; Ryan et al., 2017). Thus, the temporal filter was set to one (i.e., a temporal resolution of 30 Hz). The spacing parameter was chosen to reflect the peak spatial resolving power of a potential predator i.e., a slingjaw wrasse or coral trout, which has been estimated at ∼10–12 cycles deg−1 based on the packing density of photoreceptors. As videos were made at distances of 30 cm, we further reduced the spacing parameter to reflect spatial resolving power of 2.5 cycles deg−1 to assess the view from a predator at 1 m away, which reflects the distance in nature in which the potential predators may see a humbug (Phillips et al., 2017).

Each video clip was analysed to determine the total mean strength of motion and the mean strength of motion in 72 direction vectors, i.e., each vector was the mean across five-degree wide directional bins. We were interested in the motion cues at the edge of the humbug where the internal pattern and background pattern interact. Thus, we account for motion created by the background pattern by subtracting the motion analysis on a 600 × 600-pixel ROI of the same video frames with just the background grating. If background motion was greater than the motion of the humbug, the motion cues were treated as zero.

The total magnitude of motion cues and the magnitude of motion over direction vectors of the humbug were fit with a linear mixed-model using the lme4 package in R (Bates et al., 2015). Strength of motion was square root transformed and residuals from the models met linearity and normality assumptions. The individual fish nested in trial number was set as the random effect. Significance of the interactions were confirmed using log likelihood ratios. Motion strength was included as the response variable with direction vector, and the size and orientation of the background grating included as fixed effects with an interaction term between them. Pairwise comparisons were obtained using the pairwise method and a p-value adjustment equivalent to the Tukey test in the emmeans package (Lenth, 2022). Significance and pairwise comparisons were undertaken as above.

Anatomical measurements of visual acuity in the humbug

Three animals were euthanised with an overdose (1:2000) of methane tricaine sulfonate salt MS222 (Sigma) buffered with an equal amount of sodium bicarbonate. Retinal wholemount procedures were adapted from Ullmann et al. (2012). The retinal wholemount technique is a proven methodology for assessing retinal neurons in the eye and estimating spatial resolving power (visual acuity) across a range of species (Ullmann et al., 2012). Retinal ganglion cell counts were performed by an experienced researcher on an Olympus BX53 compound microscope fitted with a motorised stage and an Olympus DP80 camera. Stage movement and the camera was controlled by cellSens Dimension software (Olympus, version: 4.1). The total number of neurons in the retina were counted using a ×100/1.40NA oil immersion objective. Counts were made at 0.25 mm intervals with a 50 × 50 µm counting frame, providing approximately 400 sample locations across the retina. Given the difficulty in differentiating between ganglion and amacrine cells in the areas of high density, all neural cells were counted together. The theoretical (anatomical) peak spatial resolving power, expressed in cycles per degree (cpd), is a measure of the minimal separable angle at which two points can be differentiated. Spatial resolving power was estimated for the humbugs from the maximum density of neural neurons in the retina and the focal length as outlined by Lisney & Collin (2008). We assume that retinal ganglion cells are the limiting factor for spatial resolving power, and they are packed in a hexagonal array.

The effect of visual background on humbug behaviour

Humbug behaviour in response to different visual backgrounds was tested in open field trials. Open field trials are used to assay general activity and exploratory behaviour in an animal (Champagne et al., 2010). The backgrounds were large square-wave gratings of five different spatial frequencies (i.e., grating period) that were presented in two different orientations (vertical or horizontal) (see Fig. S1 for examples of the gratings). We assessed two behaviours in response to the different backgrounds: (1) the mean distance that fish positioned themselves from each different grating, and (2) the mean distanced moved throughout the tank in response to the different gratings.

Behavioural assay

Fish behaviour was tested using an open-field trial in a 40 × 22 cm aquarium. The test board (22 × 20 cm) with the background test grating was placed inside the aquarium, five cm from the short side of the tank prior to commencement of the trial making the test arena 22 × 35 cm. The aquarium was covered around the sides so that fish were unable to see outside of the tank and filled to a depth of five cm with aged water at the same temperature as the holding tanks. A water depth of five cm allowed us to record the fish behaviour on a 2D plane of motion. An air stone was placed behind the test board so that the water remained oxygenated but did not interfere with fish movement. Each of the 10 fish was tested against a total of 10 different treatment gratings, i.e., the five different spatial frequencies in both vertical and horizontal orientation. The order that each grating was presented was randomised using a pseudorandom number generator before commencement of the trials. The order of the fish to be tested against each grating was also randomised. At commencement of the first round of trials, the first treatment grating was placed into the tank. The first fish was transferred from its home tank to the middle of the experimental arena and given a 1-min acclimation period before the 5-min trial commenced. Fish were not habituated to the test arena prior to the acclimation period as it was important that we observed fish behaviour in response to each of the background gratings in a novel environment. Fish behaviour was recorded using a GoPro™ Hero 9 digital video camera positioned centrally above the tank. The GoPro camera was set to linear mode and video recorded at 30 frames per second (fps). Observers remained out of the view of the fish for the entirety of the trial (Fig. 2). At the completion of the 5-min trial, the fish was returned to its home tank and a 50% water change in the experimental arena was performed. The trials were repeated until each fish had been tested against the first treatment grating. At the completion of the first treatment, a 70% water change was undertaken and fish were given at least an hour rest before the second treatment grating commenced. This was repeated until all 10 fish had been trialed against each of the 10 different treatments. The first round of trials were undertaken over five consecutive days between 10am and 2pm. After a five-day break, a second round of trials was completed with every fish tested against every grating using the same protocol as round one, but with a different randomised order compared to the first round.

Figure 2 Experimental arena.

Schematic of the experimental arena used in the humbug behavioural trials. The observer remained out of view for the entirety of the five-minute (plus one minute acclimation) trial. The schematic was created in Adobe Illustrator.

Video tracking

The location of the fish in each video frame was obtained using the automatic tracking software DeepLabCut (DLC; version 2.2) (Mathis et al., 2018; Nath et al., 2019). Because the water level in the test arena was maintained at five cm and, therefore, the humbugs remained at a similar depth throughout the trials, only X and Y coordinates were tracked. DLC was used to track two points of interest as reference landmarks, the black stripes on the head and tail of the humbugs, but subsequently only the X, Y coordinates of the head were used for analysis. Videos were modified from 30 fps to 15 fps to reduce processing time. To train the networks to automatically track the fish, we labelled 200 frames taken from 10 videos and used a MobileNetV2.1 based neural network with default parameters for 30,000 training iterations. We validated the training algorithm with one shuffle and found the test error was: 3.2 pixels, train: 3.4 pixels. We then used a p-cutoff of 0.9 to condition the X, Y coordinates for future analysis. This network was then used to analyse the videos of all the trials which were all recorded under the same experimental settings. We obtained 4,500 positional X, Y coordinates (one per video frame) for each individual fish for a single trial. Pixels were converted to centimetres (cm/pixel = 0.02763) before calculating the distance that each fish positioned itself from each grating and the total distance travelled by each fish during a trial.

We obtained behavioural measurements every 15 s (or 225 frames). To obtain the distance that the humbugs positioned themselves from the grating, the perpendicular distance from the top of the grating to the head of the fish (i.e., grating Y –fish Y) was taken every 225 frames. For the distance moved by each fish we aggregated and summed the distance moved between coordinates every 225 frames. We chose to average the distance from the grating and the distance moved every 15 s to capture the temporal dynamics of their behavioural responses to the background conditions while balancing the need for a sufficiently fine-grained analysis. We chose 15s intervals as a suitable time to capture behavioural characteristics of reef fish as per Raoult et al. (2020).

Statistical analyses

Before investigating differences in the distance in fish position from the grating we first assessed the distribution of the data. Due to the bimodal distribution of the data, a linear model could not be fit. Because the perpendicular distance of the fish to the grating was limited to values between 0 and 35 cm due to the dimensions of the test arena, distance values were normalised between 0 and 1. Consequently, the distance data were beta distributed and a beta regression model was fitted to the data using glmmTMB in the glmmTMB package (Brooks et al., 2017). We constructed two models, the first included grating orientation (horizontal or vertical), grating size and trial time as fixed factors with a three-way interaction term between them. The second model included grating orientation and grating size with a two-way interaction term between them. We included a random effect of fish nested in round in both models. To check for overdispersion we used the overdisp function (Gelman & Hill, 2006). Residuals were checked with the residuals function in the DHARMa package (Hartig, 2022). Significance of each interaction term was confirmed using log likelihood ratios. Final model selection was done by comparing Akaike Information Criterion (AIC) values using the AIC function. Pairwise comparisons were obtained using the pairwise method and a p-value adjustment equivalent to the Tukey test in the emmeans package (Lenth, 2022).

To examine differences in the total distance travelled during the trial we fit two linear mixed-models using the lme4 package (Bates et al., 2015). The mean distance travelled was included as the response variable with grating size, grating orientation and trial time included as fixed effects and with a three-way interaction term between them in the first model. The second model included grating orientation and grating size with a two-way interaction term between them. The individual fish nested in round number was set as the random effect. The data were log-transformed and residuals from the models met linearity and normality assumptions. Significance of the interactions were confirmed using log likelihood ratios. Pairwise comparisons among main effects were obtained using the pairwise method and a p-value adjustment equivalent to the Tukey test in the emmeans package (Lenth, 2022) (R Core Team, 2022).

Results

Disruption to visual motion cues caused by background patterns

To understand how the size and orientation of background stripes effects motion dazzle, we compared motion cues of humbugs from the perspective of a potential predator. The strength of motion created at the edge of the humbug varied significantly between motion direction vectors depending on the size and orientation of the grating (X 2 = 656.8, P <  0.001). At the grating size of 0.05 cm the grating was not discernible by the virtual predator which is indicated by the black background (top inset Figs. 3A and 3B). As expected, there was no significant difference in motion created by vertical and horizontal gratings at the smallest grating of 0.05 cm. Most motion arises from the internal stripes of the fish, with majority of motion generated both forwards (blue) and backwards (red) in relation to the direction of the fish (bottom inset Figs. 3A and 3B).

The 0.25 cm gratings produced greater total motion than other grating sizes, but motion was only generated in a single direction, indicated by the solid background colours (Figs. 3C and 3D, top inset). Motion dazzle is depicted in 2DMD models as conflicting motion, produced in two opposing directions. The model for the 0.25 cm grating shows that these background gratings did not produce motion dazzle. However, when the background motion colour was removed from the image (bottom inset, Figs. 3C and 3D), the outline and shape of the humbug is easily identified. At 0.25 cm size gratings, motion strength was significantly different at 17 out of 72 motion directions between the horizontal and vertical gratings. Greater motion was detected by the virtual predator at the edge of the humbug in the vertical motion directions when the humbug was viewed against the vertical gratings, and vice versa in the horizontal gratings. The motion is dominated by the internal body stripes of the humbug when viewed against horizontal gratings, whereas the motion at the edges of the humbug dominates when viewed against vertical gratings.

Figure 3 Direction and strength of motion cues from the 2DMD model.

Direction and strength of motion cues from the 2DMD model. Direction and strength of motion cues over 360 degree directions of a the humbug viewed against background gratings sizes of 0.01 cm (A, B), 0.25 cm (C, D) and 1 cm (E, F). Panels (A, C, E) show motion strength for horizontal gratings and (B, D, F) for vertical gratings. Red dots indicate angle vectors that were significantly different between the horizontal and vertical grating of the same spatial size and black dots were not significantly different. Units are an arbitrary scaling value. Insets show example frames of motion direction over pixel location, where pixel colour corresponds to the motion direction in the colour wheel (inset A). Top insets show motion direction with the background motion, bottom insets show motion direction without background motion. Red colouration indicates motion in the opposite direction of the fish (270° ), blue colour shows motion in the direction that the fish is moving (90° ), yellow shows motion direction in an upwards direction (0° ) and the dark pink indicates motion in a downward direction (180° ). Black indicates no perceived motion.

At a grating size of 1 cm, 5 out of the 72 motion directions were significantly different between the horizontal and vertical gratings (Fig. 3). Similar to grating sizes of 0.25 cm greater motion was detected by the virtual predator in the vertical motion directions when the humbug was viewed against the vertical gratings, and vice versa in the horizontal gratings. The individual background stripes are easily resolved by the virtual predator (top inset, Figs. 3E and 3F). However, the vertical background grating caused a large amount of conflicting motion cues producing both motion in the direction travelled and opposing it, similar to the internal body stripes of the fish. When the background motion colour was removed from the image (bottom inset, Figs. 3E and 3F) the outline of the humbug is broken up and is more inconspicuous when viewed against the vertical gratings. Thus, vertical gratings above 0.25 cm, in which the individual gratings become discriminated by the virtual predator would be expected to make it more difficult for the predator to detect the edges of the humbug and the apparent direction of movement.

Anatomical measurements of visual acuity

Three retinas from the left eyes of fish were examined to establish visual acuity and areas of highest cell density. The mean peak retinal cell density across the three fish was 28,983 cells mm−2 and the anatomical acuity estimate from this is 2.65 cpd.

The effect of visual background spatial frequency on humbug behaviour

Distance in relation to different gratings

The three-way interaction of grating, orientation and time was not significant (X2 = 5.007, P = 0.286) (Fig. S2). The interaction with time was dropped and the final model specified an interaction between grating and orientation (AIC = −1,383.67). We did not include time as a covariate in the final model given that the AIC value was similar to the two-way interaction of grating and orientation (AIC = −1,383.63) and our primary interest was in assessing the background on fish behaviour rather than temporal trends.

There was a significant interaction between grating orientation and grating size (X2 = 36.150, P <  0.001). At the grating stripe width of 0.50 cm, fish remained significantly closer to the vertical grating (mean distance 13.5 cm) when compared with the horizontal grating (mean distance 17.7 cm; t = 7.602, P < 0.001). Similarly, when tested with a grating stripe width of 0.05 cm fish remained significantly closer to the vertical grating (mean 16.88 cm) than to the horizontal grating (mean distance 18.20 cm; t = 2.064, P = 0.0390). At the grating stripes widths of 0.10 and 0.25 and 1.00 cm there was no difference in fish distance between the vertical and horizontal orientations. When comparing within different orientations, fish moved significantly closer to the 0.10, 0.15 and 1.00 cm horizontal gratings when compared with the 0.05 and 0.50 horizontal gratings. Fish moved significantly closer to the 0.50 cm vertical grating when compared to other vertically orientated gratings (Fig. 4, see Tables S1 & S2 for all pairwise comparisons and test statistics).

Figure 4 Distance from grating.

Violin plot displaying the distribution of humbug positioning in relation to the horizontal (H) and vertical (V) gratings every 15 s. Mean distance (dot) and standard error (error bars) for humbug positioning every 15 s is also provided. Plot is based on 20 points per individual (distance taken each 15 s). An asterisk (*) indicates significant differences (P < 0.05) between horizontal gratings compared to the vertical grating of the same size. Letters indicate significant differences (P < 0.05) between gratings of the same orientation.

Distance moved in relation to different gratings

The three-way interaction of grating, orientation and time was significant (X 2 = 11.452, P = 0.021). Inspection of the movement over time showed no obvious differences in fish movement in response to the background over time (Fig. S3) and the AIC value (11,560.78) was similar to the two-way interaction model of grating × orientation (11,575.58). Further, given our primary interest was in assessing the effects of background on fish behaviour rather than assessing any temporal trends, we did not include time as a covariate in the final model.

There was a significant interaction between grating orientation and grating size (X2 = 115.461, P <  0.001). When comparing between the different orientations, fish moved significantly more in response to the 0.05, 0.10 and 1.00 cm vertical gratings when compared to the horizontal gratings of the same size. With the 0.25 cm and 0.50 cm gratings, the fish moved significantly more in response to the horizontal, rather than the vertical gratings. When comparing within different orientations, fish moved significantly less in response to the 0.05 cm horizontal grating when compared with the other gratings. There were no other differences in distance moved between the other horizontal gratings. The fish moved significantly less in response to the 0.25 and 0.5 cm vertical gratings compared to the other vertical gratings. There were no differences between distance moved for 0.05, 0.1 and 1 cm when vertically orientated (Fig. 5, see Tables S3 and S4 for pairwise comparisons).

Figure 5 Distance moved.

Violin plot displaying the distribution of distances moved in relation to the horizontal (H) and vertical (V) gratings every 15 s. Mean distance (dot) and standard error (error bars) for humbug movement every 15 s is also provided. Plot is based on 20 points per individual (total movement averaged each 15 s). An asterisk (*) indicates significant differences between horizontal gratings compared to the vertical grating of the same size (∗P < 0.05, ∗∗P < 0.005). Letters include differences between gratings of the same orientation.

Discussion

In this study, we found that the striped pattern of the humbugs can generate confusing directional visual motion cues (motion dazzle) for a moving predator. This motion dazzle is further enhanced when viewed against high contrast backgrounds which have similar spatial frequency and orientation to humbug stripes. In addition, the edges of the humbugs are also harder to detect against some backgrounds, particularly when the body stripes and background align. Furthermore, this study found humbugs can likely perceive the different visual backgrounds and potentially modify behaviour to optimise protection. Both fish proximity to the grating background and the total distance moved by the fish were influenced by grating width and orientation.

Humbug motion cues against different backgrounds

The motion dazzle effect, in which motion cues are generated in the opposing direction to animal movement, occurred from the striped humbug body pattern. Comparable with previous modelling and behavioural studies that assessed zebra patterning, the humbug stripes created similar confusing motion cues (How et al., 2020; How & Zanker, 2014). However, as a large focus of the motion dazzle analysis was to understand how high contrast backgrounds interact with body pattern to further disrupt motion cues, the discussion is largely focused on how different backgrounds may enhance motion dazzle effects.

Visual inspection of motion resulting from the background gratings close to the limits of the virtual predator’s visual system (0.25 cm), shows these gratings did not produce motion dazzle but rather created a large amount of motion in a single direction (Fig. 3D, top inset). At grating sizes larger than 0.25 cm where gratings are more easily resolved, motion dazzle is generated from both the humbug pattern as well as the background (see Fig. 3F (1 cm vertical) for example). There are few behavioral studies assessing motion dazzle using background patterns consisting of high contrast stripes, largely because they are not a particularly natural background. Most studies use more complex behavioral backgrounds, when visual noise makes identifying objects with high contrast stripes difficult, which may also occur with humbugs in the wild (Matchette, Cuthill & Scott-Samuel, 2018; Rowe et al., 2021; Stevens & Merilaita, 2011; Umeton et al., 2019; Xiao & Cuthill, 2016). Future studies should model motion cues of humbugs in nature viewed both in schools and against branching corals, which have a wider range of spatial frequencies and are more complex than the humbug body stripes. Our results suggest that motion, induced either from the movement of the humbug or the predator, may play an important role in causing motion dazzle particularly when viewed against high contrast backgrounds.

The motion analysis also revealed that vertical background gratings which are resolved by the virtual predator may have the added benefit of interrupting motion of the edge of the humbug. Visual inspection of motion on the humbug when background motion was removed (Figs. 3E & 3F, bottom inset), showed the greatest interruption to the humbug edge was at the 1 cm grating, when the humbug body stripes were aligned with the vertical background gratings. At this grating size the motion on the humbug is no longer depicted as a single connected object but multiple smaller objects, suggesting the humbug is more inconspicuous. This edge interruption is very similar to disruptive camouflage that has been reported to occur in static settings in a number of reef fishes (Castillo & Tavera, 2022). Our results suggest that with vertical gratings bigger than 0.25 cm (where individual gratings become discriminated by a predator), humbugs may benefit not only from the motion dazzle created by both internal stripes and the stripes interacting with the background, but also from intermittent edge disruption as the animal moves across this contrasted background, making it difficult to judge its trajectory.

The 2DMD models offer insight into the visual processing of motion cues at the level of the retina; however, a large amount of higher-order processing also occurs (Aptekar & Frye, 2013; Lee & Nordström, 2012). Thus, behavioural tests are required to assess predator responses to better interpret these models. Other shortcomings of the modeling include the limited spectral range of the camera and the approximated and simplified movements of the predator. The videos are limited to the human visual range, whereas many reef fishes have sensitivity outside of this spectral range (Marshall et al., 2019; Stieb et al., 2017). We also approximate the movement of the predator by placing the camera on a dolly system; however, movement patterns of fishes are far more complex than this (Satterfield, Claverie & Wainwright, 2023; Vidal et al., 2023). Despite these limitations, the modeling is a useful tool to understand mechanisms that can be further assessed in behavioural experiments.

Visual acuity estimates

The vital first step in evaluating how humbugs respond to different backgrounds was to understand their capacity to resolve the different grating sizes. The humbugs used in this study were found to have a peak anatomical visual acuity of 2.65 cycles deg−1, which is relatively low when considering that the average acuity for 159 teleost fish is 8.4 (± 6.5) cycles deg−1 (Caves, Sutton & Johnsen, 2017). However, it is likely that the functional (behavioural) acuity is even lower, as studies on other small coral reef fishes show that behavioural acuity is typically around half that estimated from anatomical measurements (Champ et al., 2014; Parker et al., 2017). Given this relationship, we have estimated here that the behavioural acuity of the humbugs is likely around 1.1 cycles deg−1. This estimate of behavioural acuity can be used to determine how far away the humbugs can potentially resolve objects or patterns. While this is fairly low, it is worth pointing out that many pomacentrids (damselfishes) have visual sensitivity in the ultraviolet (Cortesi et al., 2020) as well as excellent contrast enhancement which likely facilitates visually driven behaviours such as feeding on zooplankton (Hawryshyn et al., 2003). A behavioural acuity of 1.1 cycles deg−1 means that one just-resolvable cycle will subtend an angle of 0.909 degrees. A grating cycle is one black and one white band, thus for the 0.05 cm grating (1 cycle = 0.1 cm) the stripes should become unresolvable by the humbug at a distance greater than ∼6.25 cm. Similarly, a grating stripe width of 0.25 cm (1 cycle = 0.50 cm) would be unresolvable at a distance greater than ∼30 cm.

Behavioural modification in response to different background gratings

In this study we observed a significant difference in the position of free-swimming humbugs in response to different backgrounds. Overall, the humbugs were positioned closest to the 0.5 cm vertical grating which was expected given that the 0.5 cm stripes are similar in both size and orientation to the humbug stripes. Backgrounds that are similar in size likely provide humbugs with disruptive camouflage when still (Phillips et al., 2017) or reduced detection when moving (as described above). For the vertical gratings, the fish were positioned furthest from the 0.05 cm grating. This may be because this grating was difficult to resolve from most areas of the tank but also possibly because the background may have resembled an open or exposed habitat which poses greater predation risk. When tested in isolation in laboratory trials, common minnows (Phoxinus phoxinus) have been found to associate strongly with resolvable vertical stripes (∼4 cm width) which is thought to reflect a sheltering or refuge seeking response (Miles, Vowles & Kemp, 2021). Given the apparent lack of refuge to the humbugs, it is not surprising that fish were positioned further from the smallest gratings. In response to the horizontal gratings, the fish were positioned furthest from the 0.05 and 0.5 cm gratings. Similar to the vertical grating, at 0.05 cm the grating is both unresolvable and it is understandable that fish move further from this grating. However, at 0.5 cm where the gratings are resolvable from all areas of the tank and the widths are the most similar to the spacing of the humbug stripes, the reason for the distance is unknown. Perhaps the vertical stripes of 0.5 cm characterise the appearance of conspecifics and this particular size grating in a different orientation caused some uncertainty in the fish.

Humbugs moved a significantly greater distance overall in response to the smaller (high frequency) gratings, with the most movement observed in response to the 0.05 cm vertically orientated grating. At this higher spatial frequency, it is likely that the fish could not resolve the striped patterns while in areas of the arena far from the test grating (>30 cm away) and this led to the change in behaviour. However, the fact that the fish moved significantly more only when the gratings were oriented vertically suggests that they could resolve the pattern at some points within the tank and that this acquired knowledge created a persistent change in behaviour even in areas of the arena where it could not be resolved. The increase in movement by the fish may be a response to a perceived lack of background against which to conceal itself, leading to greater fear or exploratory behaviour in the relatively unfamiliar surrounds of the test arena. Alternatively, as demonstrated earlier in the motion modelling, the increased motion could represent a greater reliance on the internal contrast of the fish’s striped body pattern to generate disruptive motion cues (see Figs. 3A & 3B) (motion dazzle) that could momentarily confuse a potential predator in the final moments of a predatory strike. Interestingly, humbugs also moved more in response to the 1 cm vertical grating. While the difference moved between 0.25, 0.5 cm and 1 cm is statistically significant, it may be biologically meaningless because the effect size is so small (Eta2 = 0.00031). Given background motion dazzle and edge disruption was highest for gratings between 0.25 and 1 cm we would expect humbugs to move similar distances when responding to these to these gratings. Future studies which tease out whether these differences are biologically meaningful should be explored. Moreover, further research should also incorporate different age classes to assess whether fish and stripe size influences movement around different sized backgrounds.

Contrary to expectations, the fish moved significantly less in response to the vertical gratings that were most similar, or slightly smaller than their stripes (0.25 and 0.5 cm) compared with other vertical gratings and with horizontal gratings of the same size. Our motion modelling suggests that background motion dazzle and edge interruption is most effective when gratings are larger than 0.25 cm. Given that the background motion dazzle effect is driven by the movement of the predator, at background gratings above 0.25 cm remaining still may improve camouflage by reducing motion parallax as well as allowing the fish to conserve energy. Alternatively, it could be when the fish pattern is similar to the background that the fish remains still, relying on background or disruptive camouflage for protection. This is in line with the findings by Phillips et al. (2017), where disruptive camouflage and subsequent predation on static humbugs was less where the background was similar, or slightly smaller than the humbug stripes. It is likely that high contrast stripes are providing multiple benefits to animals through both disruptive camouflage and motion dazzle (Caro & Koneru, 2021; Stevens & Merilaita, 2011). Indeed, this has been demonstrated in patterned snakes which use their stripes or patterns to blend into their environment when still, but likely benefit from motion dazzle when rapidly fleeing predators (Valkonen et al., 2020; Wolf & Werner, 1994). Studies that explore how humbugs respond to different backgrounds in the presence of predators are certainly warranted.

Taken together, these observations suggest that humbugs may alter their behaviour depending on the environmental context. When detecting backgrounds with a spatial structure similar to their own striped body pattern, they move closer and reduce movement, which may enhance camouflage against the background and avoid revealing themselves to predators. But where cryptic camouflage is not attainable, they may use motion camouflage to confuse predators. Indeed, it is better not to be seen at all, but if and when required the humbugs may rely on motion dazzle to avoid capture. In their natural environment, it is likely that the humbugs use the branching coral colonies and other humbug fish to attain camouflage. However, humbugs have been shown to prioritise feeding in the water column over protection of the coral colony at high tide (Kent et al., 2019). It is possible that in this more exposed environment when they are without the coral complexity for protection and camouflage that they are more reliant on the motion dazzle afforded to them by their stripes. This may be why we observed greater humbug movement in response to the smaller gratings. The experiment here sought to assess the behavioural responses of humbugs to different backgrounds in a novel environment without extensive habituation. However, this approach did mean that the humbugs were exposed to an unnatural environment and the shallow water depth used in this study may have contributed to some changes in humbug behaviour. Furthermore, while 2D striped backgrounds provide a good foundation for initial exploration they do not represent the variety of complex structures found in the humbug’s natural habitat. Nevertheless, this study has demonstrated that humbugs may alter behaviour in response to different backgrounds, and future studies should consider exploring their behaviour in an environment which better reflects their reef habitat to get a more comprehensive picture of how humbugs attain camouflage in the wild.

Humbugs are not the only animals that modify their behaviour according to the spatial structure of the background (Stevens & Ruxton, 2019). Studies have demonstrated that shore crabs (Carcinus maenas) (Twort & Stevens, 2023), Aegean wall lizards (Podarcis erhardii) (Marshall, Philpot & Stevens, 2016) and the least killifish (Heterandria formosa) (Kjernsmo & Merilaita, 2012) prefer background habitats that help facilitate camouflage. In the case of the bark-resting moth (Jankowskia fuscaria), individuals have been found to increase camouflage after resting on tree bark by realigning and shifting their body position (Kang et al., 2015). Some other species of reef damselfish, Pomacentrus moluccenis and Chromis viridis, likely use a combination of body colouration and behaviour to communicate with conspecifics and maintain obscurity to predators (Marshall, 2000). Likewise, some ground-nesting birds can assess their degree of camouflage and predation risk and adjust their behaviour accordingly (Wilson-Aggarwal et al., 2016). In this study, we found that humbugs altered their behaviour by moving closer to backgrounds that potentially offered greater camouflage and may move more in response to less cryptic backgrounds, thereby utilising the striped pattern of their bodies to provide confusing motion cues. It has been proposed that irregular locomotion, animal orientation and erratic movement are also key in creating spurious motion signals (Cuthill, Matchette & Scott-Samuel, 2019; Hogan, Cuthill & Scott-Samuel, 2016), and it is likely that confusion is more effective when there is movement in different directions (Von Mühlenen & Müller, 1999). Future studies should investigate how animals can alter their behaviour to maximise the motion dazzle effect by broadening the scope of behaviours assessed.

Conclusions

In this study, we found that striped patterning of the humbug may be used for multiple defence strategies. This is the first study to demonstrate that humbug fish can generate motion dazzle, and that this is likely enhanced by certain backgrounds. Depending on the background stripe width and orientation, humbugs appeared to modify behaviour to maximise the protection offered via disruptive camouflage or motion dazzle. This is a baseline study, assessing uniform grating sizes in a controlled setting without consideration of ecological factors. However, nature is not uniform, and it has been suggested that motion dazzle requires movement of both prey and predator and is likely to depend on the background environment (Franklin, 2022). Several ecological factors including complexity in environment, attenuating properties of water and the presence of other similar damselfish could all influence the effects of motion dazzle. The humbug damselfish provides an accessible system with which to explore these questions of motion camouflage. It is likely that motion dazzle is not a ‘one-size-fits-all’ scenario which can lead to conflicting research findings.

Here, we found that where the background is in discriminable to a viewer the humbugs may rely on the confusing motion cues created by internal stripes but where the background is high contrast and resolvable that they can rely more on disruption of edge detection, and confusing motion cues induced by both the background and body patterning. We suggest that future studies consider motion dazzle based on three components: (1) movement of the striped animal, (2) the background environment (complexity, movement and lighting) and (3) the viewer’s visual system and capacity. However, there are possibly multiple defence strategies available to conspicuously patterned animals, which are likely driven by the environmental context. An integrated approach that combines modelling, behavioural trials and field experiments will be essential for gaining further understanding of how and when animals use these defence strategies. These findings will provide greater insights into the evolution of behaviour and colouration. This is exciting research that increases our understanding of the motion dazzle phenomenon and demonstrates the need for greater understanding of the interaction between pattern and motion.

Supplemental Information

Supplemental Information 1 Supplementary figures and tables

Supplemental Information 2 R Code for the motion modelling analysis

The R Code used to undertake analysis to assess differences between strength and direction of motion cues from the 2DMD motion modelling

Supplemental Information 3 R code - fish behaviour analysis

The R Code used to analyse differences in fish distances from different gratings and differences in distances moved in response to the various background gratings.

Supplemental Information 4 Raw data for motion analysis

This dataset is for the motion modelling that was undertaken.

Supplemental Information 5 Fish behaviour raw data

Thanks to Jason Martin-Powell and the Animal Research Staff at Macquarie University for all their assistance with fish husbandry; Jessica Toopitsin, Maria Pozo Montoro and Lou Beata for help with fish behaviourl trials; Ajay Nerandra for advice with video analysis and invaluable feedback on experimental design and Andrew Allen for guidance with statistics. We acknowledge the Wallumattagal people of the Dharug Nation who are the Traditional Custodians of the land on which we worked. We recognise their continuing connection to land, water and community. We pay respect to Elders past, present and emerging.

Additional Information and Declarations

Competing Interests

Author Contributions

Animal Ethics

Data Availability

The authors declare there are no competing interests.

Louise Tosetto conceived and designed the experiments, performed the experiments, analyzed the data, prepared figures and/or tables, and approved the final draft.

Nathan S. Hart conceived and designed the experiments, authored or reviewed drafts of the article, and approved the final draft.

Laura A. Ryan conceived and designed the experiments, performed the experiments, analyzed the data, prepared figures and/or tables, authored or reviewed drafts of the article, and approved the final draft.

The following information was supplied relating to ethical approvals (i.e., approving body and any reference numbers):

The Macquarie University Animal Ethics Committee provided full approval for this research (ARA No. 2017/039)

The following information was supplied regarding data availability:

The R code for motion analysis and behavioural analysis and raw data are available in the Supplementary Files.

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
