# Peer review of "Dazzling damselfish: investigating motion dazzle as a defence strategy in humbug damselfish (Dascyllus aruanus)"

_PeerJ, doi:10.7717/peerj.18152_

## Round 0.1 · original submission · Major Revisions

I apologize for the delay in completing my decision on your manuscript. It was the last of several manuscripts that arrived at almost the same time, so it was a little while before I had time to get started on it.

Overview
This study examined how the background affects motion dazzle in humbug damselfish using a computer model and checked whether live fish are sensitive to the orientation and scale of the background. The model provided evidence of additional benefits of the background on perception by a potential predator and indicated that humbugs did respond to the background with changes in distance from the background and in overall activity.

I am not able to judge the modeling approach because it is well outside my areas of expertise (or even familiarity), so I am relying on the reviewers for that. On the other hand, I have substantial experience with laboratory behavioral studies as well as watching (Caribbean) coral reef fishes in the field, so I can provide more detailed comments on that part of the study.

Both reviewers found the study original and interesting as did I, but we all had numerous suggestions for major revision. You may treat my comments below as a third review, make changes where the comments make sense or provide a detailed justification if you decide not to make the suggested changes. I will provide a pdf with yellow highlights to indicate minor problems with spelling, grammar, clarity or wordiness also with inserted comments to suggest replacement wording or to explain the concern. You do not have to include all these comments in your rebuttal except when you decide not to follow the suggestions.

Editor’s Comments
Introduction
L41ff. There is an organizational problem with the first two paragraphs of the Introduction. It starts with a framework of camouflage and implies that there are potential mechanisms of camouflage of moving animals. When it starts to focus on motion dazzle in the second paragraph, however, it recognizes that this is a form of pursuit disruption rather than detection disruption. Having read the manuscript, I see that the manuscript actually addresses a camouflage component, so this whole issue needs to be clarified.
L85. ‘biologically relevant experiment’ is too vague. You need to be much more precise about the knowledge gap that you are addressing.
L92-94. This seems to be the hypothesis that you are testing. Wouldn’t it be more appropriate to devote a paragraph to the empirical, theoretical, logical and historical underpinnings to this hypothesis rather than simply asserting it? Why would you think it to be the case that the background causes additional disruption? Have other researchers already made this point?
L96-101. I think that you could improve the description of your study species to help readers understand the ecological context of your model and laboratory study. Instead of ‘small’, provide the range of adult total lengths. You imply that they are group-living by referring to colonies, but don’t say so directly. Are they always in groups (contrary to your testing situation)? If so what are the group sizes? How do they respond to being alone? Although there are images of the fish in Fig. S1, I think it would be helpful to include a photo of the fish, preferably in their the natural habitat in the main text. Not all readers will have a clear idea of what Acropora and Pocillopora look like. Is anything known about predation threats? A specific description of the number and sizes of the stripes and the background color would be appropriate. You mention stripe width later in the Introduction, but specifying it here would be helpful. You use stripe width extensively in the interpretation of your findings, so this information could be of critical importance. Perhaps recognize that not all stripes are of equal size and provide a measure of white as well as black widths as a proportion of fish size as well as absolute size. Are the white bars similar in width to the black ones? Are the fish normally in a horizontal swimming position or do they frequently move to the diagonal (e.g., picking at the substrate, rising for plankton, displaying to conspecifics)?
L98. Much of the literature on fishes and snakes uses the term stripes for lines parallel to the long axis of the body and bars for lines perpendicular to the long axis. I don’t doubt that there is a precedent for calling to these stripes (zebras and tigers, for example), but I wonder whether you need to clarify or justify the term?
L106. This reference to movement seems to directly contradict the ‘rarely move’ assertion on L99. Paragraph needs clarification and reorganization.
L115-116. This statement seems a bit beyond what you did empirically. You can’t know if they know which background gives them better protection. All your experiment can do is determine whether they change their behavior with different backgrounds in a way which is likely to enhance the dazzle effect, according to your model.
L122. I wonder whether ‘spatial frequency’ is the correct term here. It seems to refer to repetitions per unit distance whereas your measure is absolute width.

Methods
L145. Do you know where the fish were collected or if they were aquarium-bred? Indicate total length mean and range.
L168. Didn’t you indicate that camouflage may not be the right term?
L173ff. I had trouble following the modeling approach, but neither reviewer seemed to have this problem, so it may reflect my lack of experience with this topic. You might ask a few colleagues to read it just to make sure it will be accessible to a diversity of readers.
L173-182. This paragraph is not clear because you have not explained where the fish was and where the background pattern was in relation to the camera. As also indicated by Reviewer 2, a reorganization of the section is required for a logical flow of information.
L259. The description of the behavioral test arena is very unclear. The arrangement of the stimulus board and the test arena could have important implications for responses of the fish and for anyone wishing to repeat the experiment, so this is critical. Although some of my comments here and on the manuscript may seem picky, it is important to realize that readers may not be native English speakers and using different terms for the same things can result in confusion.
• The test tank was 40 x 22 cm (L259) and the test board was 25 x 25 (L165, assuming that background card and test board are different terms that refer to the same thing). Even though the image would apparently not fit on the short side of the tank, you should specify which side it was parallel to and also that it was inside the tank rather than outside (as I assumed on first reading). Fig. 1 does not help because it shows the test board on the short side, and the tank dimensions do not match a 40 x 22 cm tank (ratio is 1.2 rather than 1.8). On the other hand, the distance categories (L300) suggest that the board was on the short side after all. It seems like there could have been room for a humbug to move behind the board, but this is not apparent in the figure. On the other hand, if the board was more than 25 cm long, there was an area that did not have the stimulus lines; what was the color and texture of that?
• Why are the cards so big if the water was only 5 cm deep?
• Fig. 1 needs to be redrawn to be of any use to readers. It should be drawn to scale with the scale indicated. The perspective is inconsistent and distorted. For example, the test board is drawn with perspective but the black and white lines are not. It is not clear that the board is 5 cm from the wall. The fish is a crude image which appears to be from the side (?), although the tank is drawn from above and the fish seems to have both eyes on the same side, like a flatfish.
• Fig. 1 shows an experimenter’s position very close to the tank, yet the text says that observers remained out of view. Was there a blind? Was the tank opaque or with clear walls like normal aquaria? If clear, what could the fish see outside the walls?
• The water depth of 5 cm strikes me as extremely shallow. Although you have not given the fish sizes, I imagine that it might be only two or three times the height of the fish. I can appreciate that you wanted a two-dimensional image in focus but I have concerns about this. Do humbugs in nature ever experience such shallow water? Did your experimental fish ever experience it before the test? Is it possible that such shallow water triggers an escape response (move away from home range to deeper water during an extreme low tide event)?
• The protocol is not completely clear. It seems as though each individual fish was subjected to all 10 patterns. Was this done by immediately replacing one board with another after a trial, waiting only 1 min and then recording for 5 min until all 10 were presented, with a replicate of the 10 in different order 5 days later? Some of this is not clearly stated. In particular, you use ‘trial’ to refer to an individual 5-min treatment (L265) but later (L268) state that the fish was returned to its home tank followed by a partial change of water when the trial was completed. If each individual treatment was separate by removal of the fish and a water change, you need to indicate the time between treatments within a trial. This ambiguity also makes the statistical analysis unclear (L311, 322) where individual is nested within trial (which does not make sense to me if trial refers to an individual treatment).
L264. I took this to mean that 1 min of acclimation was followed by 5 min of recording. However, Fig. S2 seems to have only 4 min of data. Be clearer here about the duration of actual recording.
L298. Measuring distance moved at only 15 s intervals seems quite crude for an active fish. Can you justify this choice?
L304. Regarding the statistical analysis, one of the reviewers raised concerns that the number of fish used in the behavioral study was only 6. From my experience with behavioral studies, I would agree with this concern. You undertook a sophisticated analysis which I am not qualified to judge. However, I would like you to check with a statistician to confirm that there is no problem with pseudoreplication/repeated measures because you have some very low probabilities with large numbers of repeated measures (e.g., d.f. = 2,386 in Tables S1,S3) for small numbers of individuals.
L310. What is the measure of time? Is it within a trial or among all trials in a set?

Results
Model
L334ff. As with the Methods, I did have trouble following this section. Part of the problem may be my lack of familiarity with this approach but I presume that this approach is unique to your study or to yours and the zebra study. You may need to provide more information to guide the reader through your understanding of the patterns on the figure. I suggest that after revision, you ask some colleagues to read this section and see how well they understand what you are trying to communicate.
L339. Additional confusion: Fig. 2A,B indicates 0.1 cm but the text and Methods L178 indicate 0.05 cm.

Behavior
L382ff. I presume you used medians and boxplots in Fig. 3 because of the non-normal (bimodal) distribution. It is therefore inappropriate to refer to the means in the text. In addition, it means that the figure is inconsistent with the values in the text. Why not be consistent and use medians in both the text and the figure?
Is it really possible that 0.05 and 0.1 gratings in horizontal format are significantly different when the medians are almost identical? If so, some explanation may help.
It is quite striking that in the horizontal position and two smallest sizes in the vertical, the humbugs are far from uniformly distributed in the tank. They are almost as far from the board with the gratings as they can get. When presented with the coarser vertical bars, they move closer, but only with 0.5 are they about evenly distributed between the board and the far wall. Consider noting this in the results and considering whether it might indicate some artifact of the testing system when you look critically at your findings in the Discussion. (I see from the Discussion that this is partially addressed, but I think it needs attention in both Results and Discussion.)
Fig. 3. Formatting of the figure could be improved. You don’t need the axis label and numbers for the right hand box as it is the same as the left hand one. I suggest raising the upper frame of the box and writing out horizontal and vertical above the letters indicating significant differences. I am not sure about PeerJ, but some journals want the same level of precision for all numbers on the axis; yours range from integers to hundredths of a cm. The caption needs more information. There is more than one format for boxplots, so you should be specific about what the horizontal line, limits of box and vertical lines represent. You should indicate sample size for calculating the boxes. Is it based on means from two trials for 6 fish (n = 12, which seems small for a valid boxplot) or did you consider some larger number such as 4 distances per minute for 5 minutes for 2 trials on 6 fish? It would be advisable to ask a statistician whether whichever approach you used is valid.
Fig. S2. Improve the caption to this figure with more information as for Fig. 3. Is the parametric measure of distribution valid here or should you still be using a boxplot approach? Move the key to the empty spot to the right of panel 1.0 and describe it in the caption. Why is the x-axis labelled time point (a rather vague term) rather than minutes?
Figs. 4, S3. Please follow the suggestions for Figs. 3, S2. The y-axis label could be simplified to ‘Distance moved per 15 s (cm)’. Figure 4 does not work well. The large number of outliers means forces the median and box to be so compressed that it is not easy to tell the difference. You may have to look at other publications to see the easiest way to break the y-axis to allow the central data to be clearer.
L401ff. I think there may be a clearer and more appropriate way to describe the result here. It seems to me that for all horizontal gratings, the humbugs averaged little movement (I can’t see how small from the graphs). 0.05 may have been significantly less, but the absolute amount is quite small. For 0.25, 0.5 and 1 vertical, movement was similar and low, on average. For 0.1 vertical, movement was slightly higher and only for 0.05 vertical was it substantially higher. This focusses on the overall pattern of effect instead of only the direction and statistical significance of the difference. This makes it clearer that what needs explaining is the higher rate of movement at small vertical gratings. From this perspective, Fig. S3 may be much more revealing of the pattern that the current Fig. 4. Consider putting S3 in the text instead of Fig. 4.
Fig. 5. The caption says mean +/- SE, but the figure seems to show boxplots. Make changes as for the other figures, especially decompressing the boxplots.
Tables S1, S2. Provide more complete headings for these tables. All abbreviations should be defined and the test used should be stated.

Discussion
L428-437. This section is more like an Abstract than the start of a rigorous Discussion. It involves interpretation of your empirical findings and is not appropriate in the lead position. It would be appropriate to briefly review the empirical results before you dive into their interpretation. However, it would be easier for readers if you did this separately for each of the three sub-sections of your study.
L440-442. You need to be much more explicit about what evidence from the model supports this conclusion and address any inconsistencies in the data as well as the validity of the model.
L445. Is How & Zanker the only empirical evidence for this effect? This is where you should briefly put your study in the literature context, even if you intend to focus on the effect of the background.
L449-452. I did not understand what empirical evidence supported this statement, so I went back to the results. That did not help me. Here you write explicitly about dazzle whereas in the Results, the references are to motion. It is likely that you are so familiar with the system that you don’t recognize the challenge for new readers to grasp the concepts and evidence. Please revise the Results to make certain that the evidence is clear. Please revise the Discussion to make sure how you interpret that evidence and any ambiguities and uncertainties is clear.
L462-464. Again explain how the empirical evidence supports this interpretation.
L468. Be more explicit and explain the interpretation.
L480. Do you think you should add anything about the validity of the model and improvements that might be needed?
L486. Is it surprising that a plankton feeder has such low visual acuity?
L491. I would expect a bit more critical discussion of the reliability of this measure. Is the counting process and calculation very straightforward and unambiguous, or could it depend on experience, histological technique, sample size or other issues?
L499ff. Don’t you think you should include a bit more critical discussion of your experimental protocol and how it might have influenced the results? Issues might include extremely shallow water, position and size of the grating board, very short time for the fish to adjust to conditions in the tank after transfer, use of space in the tank beyond the mean (bimodality mentioned in Methods, for example, where the mean does not reflect where the most time was spent). I do not intend that you ‘apologize’ for all the protocol decisions but that you thoughtfully consider, based on your experience in actually running the experiment and watching the videos, factors that might have affected the results/interpretation.
L494. Is this an error? For 0.5 cm width, isn’t one cycle 1 cm, not 0.1?
L510. In looking at Fig. 3, I was struck by how far, even the closest median position was (only about the middle of the tank). Could this insight also explain that?
L517-518. Incomplete comparison. Did they move less compared to horizontal grating or compared to other vertical gratings or both? I feel that the emphasis in this paragraph on less movement with wider gratings somewhat mischaracterizes your findings. Looking at Fig. S3, it is clear that at 0.25 and 0.5, as well as 1.0, lower motion, even though statistically significant, may be biologically meaningless because the effect size is so small. Indeed, most of the dots for the means overlap. More striking to me is the similar levels of low movement at 1.0, 0.5, and 0.25, whether grating was vertical or horizontal, as well as 0.05 with horizontal grating, but substantially higher levels of movement at 0.05 with vertical grating. (It is less clear what is happening with 0.1 because both treatments seem to show more movement at some times during the trials.) I wonder if your discussion the greater movement of fish in the 0.05 vertical trials that they apparently cannot resolve should be moved ahead of this paragraph. I think that the Abstract statement about lower movement when background is similar to humbug stripe size (L33) is misleading and should be corrected. The issue is the unpredicted higher activity at 0.05. There should also be some attention to the potential for artifacts arising from the test situation.
L534. It appears that humbugs did not modify their movement rates when they were at different distances. I think this needs to be explicit as well as briefly considering how it affects your potential explanation of the higher movement at vertical 0.05.
L548. But note that the greater movement at 1.0 was much, much less than that at 0.05.
L557-559. We can’t infer what humbugs understand. We can only say that they have a behavioral to different backgrounds. We can’t even say that it is an evolved response without knowing the developmental history of the tested individuals. Depending on the situation, they could have learned a response.

Conclusions
L598. Also, worth considering ontogeny, a what size stripes develop and how fish adjust to changes in their own size?

References
Generally well done, but some journal article titles have capital letters throughout.

Reviewer 1 ·

Basic reporting

This manuscript by Tosetto et al. investigated the antipredator benefits of striped body patterns in humbug damselfish (Dascyllus aruanus) through a combination of modeling and behavioral experiments. The authors assessed whether humbug coloration functions according to the principles of motion dazzle and edge disruption. Using a 2DMD model, the authors demonstrate that motion signals produced by body stripes might provide misleading motion cues when observed against a homogeneous background. However, in the background with ‘high contrast’, the authors found that internal stripes may help disrupt edge detection. In the behavioral experiments, the authors found D.aruanus modified their behavior in response to the background. Specifically, in environments where the frequency of background stripes closely matches that of the humbug's body pattern, the fish exhibited reduced movement and tended to remain in close proximity to the background. Conversely, when the background featured a higher stripe frequency, the fish exhibited increased movement. The authors conclude that striped animals may modify their behavior to maximize protection through motion dazzle in different backgrounds.

Overall, this is an interesting manuscript that tries to test the principles of motion dazzle in a real animal. This is a positive shift, given that the majority of previous studies on motion dazzle involved virtual predation experiments with humans. I commend the author's effort for integrating behavioral experiments with visual motion modeling. However, I have substantial concerns about the concepts examined and cannot recommend the manuscript for publication in its current form. I hope the authors find these comments helpful.

I must state that my knowledge in the field of anatomical measurement of visual acuity is limited. Consequently, I cannot assess the validity of the methods and results presented.

Experimental design

1. The number of fish individuals used in behavior experiments is insufficient. First, the authors do not provide the exact number of individuals used in the behavior experiment. Second, they also do not state the number of replicates per individual. Despite the authors' randomization of trials, the use of only six individuals (the maximum number reported initially) is a notably small number. While this is not a major issue for motion modeling, I am not sure the sample size is large enough to have any conclusive results in behavior experiments. I would like to hear from authors for using only a few individuals. A strong justification must be provided in any case.

2. The experimental procedures/setup could be better reported (see also specific comments below). E.g., in lines 173-182: How big was the experimental arena? Was the experimental background shown only on a single side of the setup? What measures did the authors take to account for the distance between the background, the studied individual, and the camera? How did the authors ensure that the distance between the background and the studied individual was constant in all six presentations? How was the lighting controlled? Were there any apparent reflections? Please add details.

Validity of the findings

1. In multiple places, the authors state that they found support for ‘disruption of edge detection’. This could be easily confused with disruptive coloration that is suggested to work when the animal is not in motion. I suggest authors carefully go over the entire manuscript and make this distinction clearer (see specific comments below).

2. Complex backgrounds have high visual noise (low signal-to-noise ratio) that imposes strong visual processing costs on the predators. It is possible that in nature, the studied animals might be exploiting the visual complexity of corals to enhance concealment. However, in the context of current experiments, I am not sure if the backgrounds used by authors can be considered complex (highly repetitive grating pattern). I suggest that the authors refrain from using the term "complexity" in the title and throughout the manuscript.

3. The authors found some interesting results regarding the orientation of grating, i.e., vertical versus horizontal. However, they never discuss the implications of these findings. For example, see the paper below suggesting that the orientation of stripes may have different motion dazzle effects. I suggest that authors highlight this aspect of results in the abstract and also have section about this in the discussion.
Hughes, A.E., Magor-Elliott, R.S. and Stevens, M., 2015. The role of stripe orientation in target capture success. Frontiers in zoology, 12, pp.1-12.

Additional comments

1. Lines 19-22: This definition of motion dazzle is incorrect, especially the part about ‘reduce likelihood of identification’. Please see below a recent paper that provides a clear definition of motion dazzle. Rewrite.
Scott-Samuel, N.E., Caro, T., Matchette, S.R. and Cuthill, I.C., 2023. Dazzle: surface patterns that impede interception. Biological Journal of the Linnean Society, p.blad075.

2. Line 29: Please state clearly if the disruption is during motion or when stationary.

3. Line 36: I'm sorry, this is confusing. Shouldn't this be motion dazzle instead? Please avoid using the term ‘disruption’ here. It could easily be confused with ‘disruptive coloration.’

4. Line 48 & 52: Another recent review on the topic.
Tan, M., Zhang, S., Stevens, M., Li, D. and Tan, E.J., 2024. Antipredator defences in motion: animals reduce predation risks by concealing or misleading motion signals. Biological Reviews.

5. Line 52-54: I would argue the opposite. See the papers below that show support for these strategies in real animals, but evidence for motion dazzle is still lacking (see also a new review).
Mizutani, A., Chahl, J.S. and Srinivasan, M.V., 2003. Motion camouflage in dragonflies. Nature, 423(6940), pp.604-604.
Kane, S.A. and Zamani, M., 2014. Falcons pursue prey using visual motion cues: new perspectives from animal-borne cameras. Journal of Experimental Biology, 217(2), pp.225-234.
Valkonen, J.K., Vakkila, A., Pesari, S., Tuominen, L. and Mappes, J., 2020. Protective coloration of European vipers throughout the predation sequence. Animal behaviour, 164, pp.99-104.
Scott-Samuel, N.E., Caro, T., Matchette, S.R. and Cuthill, I.C., 2023. Dazzle: surface patterns that impede interception. Biological Journal of the Linnean Society, p.blad075.

6. Line 60: Consider citing this recent review paper here.
Scott-Samuel, N.E., Caro, T., Matchette, S.R. and Cuthill, I.C., 2023. Dazzle: surface patterns that impede interception. Biological Journal of the Linnean Society, p.blad075.

7. Line 84: This should be Murali and Kodandaramiah (2018), not Murali and Kodandaramiah (2020).
Murali, G. and Kodandaramaiah, U., 2018. Body size and evolution of motion dazzle coloration in lizards. Behavioral Ecology, 29(1), pp.79-86.

8. See also below experimental studies that used real predators.
Zlotnik, S., Darnell, G.M. and Bernal, X.E., 2018. Anuran predators overcome visual illusion: dazzle coloration does not protect moving prey. Animal Cognition, 21(5), pp.729-733.
Hämäläinen, L., Valkonen, J., Mappes, J. and Rojas, B., 2015. Visual illusions in predator–prey interactions: birds find moving patterned prey harder to catch. Animal Cognition, 18, pp.1059-1068.

9. Line 98: Are these stripes parallel or perpendicular to the anterior-posterior axis?

10. Line 109 -111 & 114-115: This sentence needs some citation. I am not sure if any studies have directly tested this. The only relevant study could be by Hughes et al. 2021. If this is a hypothesis, it must be stated clearly.
Hughes, A.E., Griffiths, D., Troscianko, J. and Kelley, L.A., 2021. The evolution of patterning during movement in a large-scale citizen science game. Proceedings of the Royal Society B, 288(1942), p.20202823.

11. Line 119: Consider adding background information about the motion models.

12. Line 128: To be clear, add ‘stationary’ after camouflage.

13. Line 138: Why only vertical orientation? Please provide some reasoning (perhaps to match the orientation of body stripes?).

14. Line 145: The number of fish individuals used is too small. Provide some justifications here. Importantly, are there any notable variations (i.e., polymorphism) in color patterns in this fish species?

15. Line 161: Consider providing these values in visual degrees.

16. Line 170: The authors should explain why they used a 2-dimensional experimental setup, considering that animals often navigate through three-dimensional environments in nature.

17. Line 181: Should there be ‘each’ before fish? If not, add the number of replications for each fish individual.

18. Line 187-189: I see the advantage of this method (i.e., moving the camera while the animal is still). However, it might take away the natural variations in motion (e.g., the type of movement may have some influence on the effectiveness of stripes). For instance, fish that move by anguilliform motion might have a different effect than thunniform motion. I think the authors should provide some details about the type of motion exhibited by the study species and acknowledge this limitation.

19. Line 187-189: I would also consider adding details about the angle subtended by the camera motion in the analysis and add discussion on how this might impact the results (e.g., body stripes may align differently with the background at different angles). Importantly, for the analysis comparing motion signal strength between background and humbug, was the 600 × 600 pixel region in the center of the arena? Ideally, the angle subtended here (i.e., background-only analysis) should be the same as the humbug’s position.

20. Line 206: Do the potential predators possess UV sensitivity? To what degree are the models sensitive to wavelengths outside the visible range (i.e., ones measured here)?

21. Line 207: 30 cm between the camera and the background? Please clarify.

22. Line 223: add ‘with’ after fit.

23. Line 234: How many animals?

24. Line 258 onwards: It is unclear how many animals the authors used in this experiment or the number of replicates per orientation-grating spatial frequency combinations.

25. Line 205: typo ‘tfihe’

26. Line 310: I may have missed it. Unclear what the variable ‘time’ means. Please provide some extra details.

27. Line 273: ‘set of trials’ How many trails before the five-day break per fish? Please provide more details.

28. Line 320: I assumed this must have been the total distance traveled per fish per trial. Please add details about how this ‘mean’ distance was calculated.

29. Line 334: remove ‘the’ before ‘how’.

30. Line 378: I presume this model also included the individual effect of grating and orientation. Please make this clearer.

31. Line 439 onwards: The backgrounds used by authors were unnatural. I suggest that the authors acknowledge this limitation in discussion as well (e.g., from lines 477).

32. Line 447,453, 470: Background complexity or complex background is something else in camouflage research (see below paper). Complex backgrounds have high visual noise (low signal-to-noise ratio). I am not sure if the backgrounds used can be considered complex. I suggest the authors rewrite this section (see main comment).
Xiao, F. and Cuthill, I.C., 2016. Background complexity and the detectability of camouflaged targets by birds and humans. Proceedings of the Royal Society B: Biological Sciences, 283(1838), p.20161527.

33. Line 448: Insert ‘frequency’ between gratings and close.

34. Line 456: There is some confusion here. The study cited is about flicker fusion, not motion dazzle. Please clarify this in the text.

35. Line 460 onwards: It is unclear how authors can draw inferences on detectability when they only measured motion signals in their 2DMD models. Please add details about the analysis the authors are referring to.

36. Line 463-467: This entire section about disruption is confusing. First, I thought that the authors meant disruption of motion signals when they write ‘disrupting detection of edge’. But here, they cite papers on disruptive coloration, which previously has been proposed to work when the animal remains still. I suggest that the authors rewrite this section to improve clarity.

37. Line 531: A relevant study.
Valkonen, J.K., Vakkila, A., Pesari, S., Tuominen, L. and Mappes, J., 2020. Protective coloration of European vipers throughout the predation sequence. Animal behaviour, 164, pp.99-104.

38. Line 557: I would rewrite this sentence as ‘these observations suggest that humbugs have evolved an ability to modify their behavior according to the backgrounds to enhance camouflage.’

39. Line 560: ‘to remain cryptically camouflaged’‒ I suggest toning this sentence down. At this stage, this is very speculative. The authors have not conducted any experiment to determine how the degree of crypsis varies with different backgrounds (i.e., survivorship of fish when viewed against different backgrounds).

40. Line 592-594: I suggest toning this sentence down. The results are based on visual models and not real animals. I would avoid making strong inferences out of it.

Reviewer 2 ·

Basic reporting

This is an important study to understand motion dazzle, as there are few studies thus far on motion dazzle. The manuscript is well-written in general, however, there are several crucial areas in the manuscript that require clarification before this manuscript can be accepted, particularly in the Materials and Methods. These areas are crucial for the analyses and interpretation of the experiment and results. Specific comments follow below.


Abstract

Lines 27-30: This sentence is too long and confusing; use of ‘that’ three times

Line 33: What are the humbugs moving closer to?

Introduction

Line 82: Motion dazzle patterns are associated with the evolution of smaller body size in which animals?

Experimental design

Materials and methods
The materials and methods needs further reorganisation to improve logical flow. The presentation of the estimation of motion dazzle includes analyses of the videos (Lines 184-210), but until this point, the reader does not know how many videos were taken and what treatments there are. Readers can only surmise that this missing information is in lines 249-273. However, this is then followed by further details on video analyses in lines 275-301.

Line 166: Would the use of the laminated pouches affect the humbugs’ perception of the test backgrounds?

Lines 173-178; Figure 1: It is not clear where the position of the ‘predator’ is; and/or where the recording positions are. According to the text, video footage was recorded from the perspective of a moving predator, but this is not represented in Figure 1. Instead, there is a ‘camera’ and ‘experimenter’ indicated in Fig 1.

Line 191: ‘Between 8 and 23 frames were analysed.’ – for each video? For the entire dataset? Or a subset? Please be specific.

Line 234: How many animals were euthanised? All six fish?

Line 249: What are open field trials?

Line 265: It is not clear if each trial refers to each presentation of background, or does that refer to the entire period of presentation of the 10 backgrounds. If the latter, for how long was each background presented to the test subject? In addition, how long was the interval between different background presentation?

Line 282: To reduce processing time, videos were modified from 30fps to 15fps. Would this affect the interpretation from the predator’s perspective?


Line 303, Statistical analyses: it is not clear how the repeated measures (re: lines 272-273) of the same humbug individual were taken into account in the analyses.

Line 305: ‘tfihe’ the?

Validity of the findings

Results

Line 345: ‘The 0.25 cm gratings produced greater total motion.’ This sentence is incomplete.

Lines 351, 358: ‘greater motion was produced in’… does this refer to greater motion of the humbug, or the predator perception of the humbug?


Discussion

Lines 471-473: ‘…no heightened capture of humbugs’ compared to?

Lines 505-506: what is the distance(s) involved?

Lines 520-522: Confusing, please rephrase.

Line 604: ‘Here we found where the background is more uniform’ as far as I can tell, all the test backgrounds are uniform. What is the comparison here?

Additional comments

Figure 1
Please include arena dimensions

Figure 2
To avoid confusion, drop ‘open’ from ‘red open dots’ since there are no red closed dots in the figure.
At the current size and resolution, the insets are too small for the colours within the fish body to be discernible and interpreted. Please increase the inset sizes and resolution or remove the insets.

Figure 5
Placement of the asterisk above the bars is not consistent; at times above the vertical bars while other times above the horizontal bars. Since ‘*Indicate significant differences between horizontal gratings compared to the vertical grating of the same size’, shouldn’t the asterisk be placed above but between the two bars?

---

## Round 0.2 · Minor Revisions

Again, I must apologize for the delay in evaluating your manuscript. It arrived at nearly the same time as others when I was already quite busy with other tasks. I checked the manuscript and decided that your responses to the comments from the reviewers and me were sufficiently clear and complete that I did not need to send it out to the reviewers for a second read and then put it aside until my desk was a bit clearer.

I found your changes have greatly improved the manuscript. Although I still have some trouble understanding the model results and interpretation, my sense is that more knowledgeable readers would be able to understand them. The behavioural design and interpretation are much clearer.

Unfortunately, there are a large number of errors in grammar and word choice throughout the manuscript. I have indicated these on the pdf, but it would be a good idea to have another reader with strong language skills take a careful look because I almost certainly missed some.

In addition, there are a few specific issues, all minor, involving the presentation that need to be resolved. Therefore, I am returning the manuscript with a recommendation of minor revisions. It should not take you long to make these changes and resubmit.

Specific comments

Fig. 4. H and V should be defined in the caption and the letters should be moved closer to the appropriate panels (V is closer to the left panel than to the right one). Describe the violin plots so that the reader understands the dot, lines and width. Also, I am not sure that the caption and y-axis label are correct in referring to the ‘mean distance’. Mean distance would be based on one point for each fish (i.e., its mean) meaning that the violin plots would be based on only 10 points. I am not sure that a violin plot based on this small sample would be possible or relevant. If you used all the points for each fish, you would have 200 in total which might be possible, although there are issues of patterns within and between individuals. I think it would help readers if you made the sample size on which these plots are based explicit in the caption.

Fig. 5 requires similar adjustments to Fig. 4. In this case, the axis seems to indicate that the plot is based on 20 points per individual (distance measured each 15 s) but the caption refers to mean and SE whereas the plot shows the distribution of all the points.

It is not clear why Table S5 is highlighted in yellow.

---

## Round 0.3 · accepted · Accept

The authors have responded appropriately to all editor suggestions, including having additional readers assist in checking grammar and clarity as well as improving figure captions. The manuscript is now ready for the publication process. The authors may want to consider adding the additional readers to their acknowledgements.